# CubistMerge: Spatial-Preserving Token Merging For Diverse ViT Backbones

## Abstract

We introduce a simple yet effective token merging method for ViTs that is compatible with modern spatial ViT architectures like SAM or DINOv3, by maintaining spatial integrity of merged tokens. Our proposal reconciles two seemingly conflicting requirements: (i) exploiting the uneven information distribution across the spatial layout while (ii) preserving the spatial structure post-merging. Our approach employs (i) a 2D reduction strategy to enforce structured token layouts, (ii) a spatial-aware merging algorithm that maintains relative token positions, and (iii) a novel max-magnitude-per-dimension token representation that preserves salient features. Our method demonstrates strong performance both off-the-shelf and with fine-tuning, achieving state-of-the-art results on spatial and non-spatial architectures across various vision tasks. Specifically, we achieve 1.25× speedup on SAM-H with only 0.7% mIOU drop evaluated on COCO off-the-shelf, and 1.15× speedup on DeiT-B with no top-1 accuracy drop on ImageNet within just one epoch of fine-tuning.

## 1 Introduction

Vision Transformers have become the leading architecture across various vision tasks such as classification (Dosovitskiy et al., 2021; Touvron et al., 2021; He et al., 2022), object detection (Li et al., 2022a; Ryali et al., 2023; Cheng et al., 2022) and semantic segmentation (Kirillov et al., 2023; Ravi et al., 2025; Strudel et al., 2021). However, their memory and computational demands pose major challenges, especially with the growing sizes of recent models (Siméoni et al., 2025).

Token reduction methods offer an attractive solution by leveraging the input-agnostic nature of transformers to dynamically reduce the number of tokens during processing. However, the vast majority of existing token reduction methods face fundamental incompatibilities with spatial architectures, such as 2D positional embeddings (Li et al., 2022b; Heo et al., 2024) at every attention layer, and window attention (Liu et al., 2021; Li et al., 2022a). Most techniques (Rao et al., 2021; Liang et al., 2022b; Kong et al., 2022; Bolya et al., 2023; Lee et al., 2024a; Tran et al., 2024; Long et al., 2023; Kim et al., 2024; Norouzi et al., 2024; Chen et al., 2023) produce unstructured token layouts that break spatial coherence (see Figure 1). The resulting unstructured token layouts break both window attention, which requires consistent token counts across all windows, as well as 2D positional embeddings, which depend on structured arrangements to compute spatial relationships correctly. The impact of breaking spatial coherence is shown in Figure 2(b): non-spatial-preserving methods like ToMe severely distort attention patterns of models with relative positional bias. In contrast, our spatial-preserving approach maintain attention patterns that closely resemble the baseline model, as shown in Figure 2(a).

Expedite (Liang et al., 2022a) is the only existing method that maintains spatial integrity, doing so by pooling across the structured feature map to initialize cluster centroids. However, the resulting clusters are distributed evenly across the feature maps, without regard to information density variation across different regions; this causes information loss and significant performance degradation (see Section 4.1).

In this paper, we show how to reconcile two seemingly conflicting requirements: (i) preserving spatial structure of merged tokens while (ii) exploiting the uneven information distribution across the spatial layout. We propose CubistMerge, a spatial-preserving token merging method that selectively joins redundant tokens using an information-preserving representation, while leaving distinct tokens

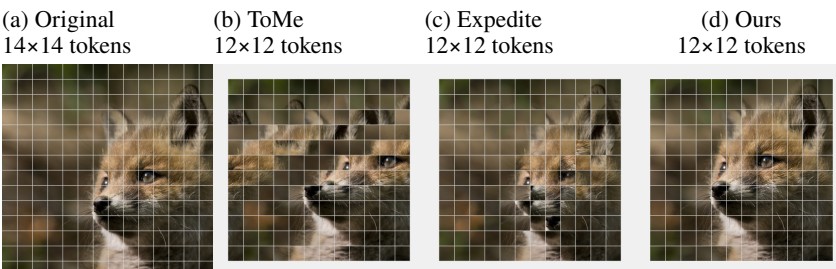

| (a) Original | (b) ToMe | (c) Expedite | (d) Ours |
| 14×14 tokens | 12×12 tokens | 12×12 tokens | 12×12 tokens |

Figure 1: (a→b): Most token merging methods, like ToMe shown here, fail to preserve spatial layouts. (a→c): Expedite preserves spatial structure, but fails to exploit information density unevenness across regions, losing information. (a→d): CubistMerge preserves spatial coherence while focusing token reduction on regions with low information density.

untouched. Our 2D reduction strategy maintains structured spatial token layouts in the resulting tokens after merging, enabling compatibility with spatial architectures. CubistMerge can operate as an off-the-shelf solution and also shows strong fine-tuning performance within a small number of epochs.

The key contributions we make in this paper are:

- a 2D token reduction strategy that maintains consistent token counts per row and column;
- a spatial-aware token merging that maintains relative spatial relationships; and
- a max-magnitude-per-dimension token representation that preserves salient features without requiring layer-wise attention rescaling.

We demonstrate generalizability through comprehensive evaluation across several influential ViT backbones on diverse tasks including classification, detection, and segmentation. On spatial architectures, we consistently outperform Expedite (Liang et al., 2022a) across every task and model. To compare against existing methods more broadly, we also conducted experiments on vanilla ViT backbones, achieving state-of-the-art results, both off-the-shelf and with fine-tuning, on DeiT-B compared against 5 other token reduction techniques. Notably, we achieve no accuracy loss on ImageNet at 1.15× speedup within just one epoch of fine-tuning. Even compared against specialized methods like ALGM (Norouzi et al., 2024) which targets segmentation, we achieve similar performance while maintaining broader applicability.

## 2 RELATED WORK

**Token Pruning.** Early token reduction methods primarily focused on token pruning (Rao et al., 2021; Liang et al., 2022b; Kong et al., 2022). While effective for early classification models, these approaches suffer from critical limitations: (1) they cannot recover discarded tokens, rendering them incompatible with modern backbones that require dense token layouts at the output (Li et al., 2022b;a; Ryali et al., 2023), and (2) they introduce extra learned parameters, necessitating retraining of additional modules alongside the backbone model.

**Retraining-Based Token Reduction.** Some token reduction approaches require extensive retraining (Long et al., 2023; Lu et al., 2023; Liu et al., 2024b; Lee & Hong, 2024; Liu et al., 2024a), which presents challenges for modern large-scale models due to computational costs of training and limited data availability: foundation models such as DINOv3 (Siméoni et al., 2025) rely on massive datasets and scale architectures up to 7B parameters. While effective, these retraining-based approaches are prohibitively expensive, creating a need for training-free solutions.

**Graph-based Token Merging.** Token Merging (ToMe) (Bolya et al., 2023) addresses both limitations above: it merges tokens rather than discarding them, enabling recovery for dense outputs, and can operate off-the-shelf without retraining. ToMe employs a graph-based approach with bipartite matching to selectively combine similar tokens through weighted averaging. This approach demonstrates success across several models and tasks, becoming the foundation for subsequent works with

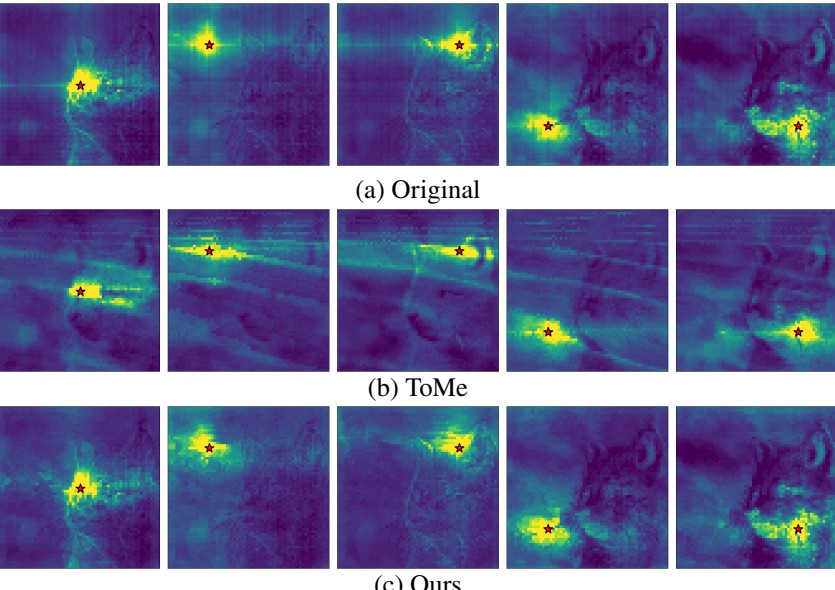

Figure 2: Attention patterns with relative positional bias towards 5 different token positions (indicated by red star) on SAM-B. (a) shows attention map of baseline model. (b) shows effective attention pattern with ToMe applied (c) shows effective attention pattern with our method applied. Our method preserves attention patterns better than non-spatial-preserving method like ToMe.

incremental improvements such as adaptive merging rates (Norouzi et al., 2024; Chen et al., 2023), importance-based token selection (Lee et al., 2024a; Tran et al., 2024; Long et al., 2023), and hybrid pruning-merging approaches (Kim et al., 2024). However, ToMe and these subsequent works all fail to maintain spatial structure after merging, which is critical for architectures with spatial components. Despite some works adopting spatial-aware strategies (Norouzi et al., 2024; Xu et al., 2024), they only focus on merging spatially near tokens but do not maintain structured spatial layouts in resulting tokens.

**Clustering-based Token Merging.** Expedite (Liang et al., 2022a) represents the only existing method that preserves spatial structure by employing a k-means clustering approach on superpixels initialized through adaptive average pooling, producing structured 2D layouts compatible with spatial architectures. However, Expedite fails to exploit information density unevenness across feature maps, and consequently fails to preserve semantically distinct tokens. The information loss on distinct tokens leads to performance drops especially when applied to early layers (see Section 4.1). AiluRus (Li et al., 2023) also noted this weakness and built upon Expedite's clustering approach, addressing this limitation by identifying cluster centers based on semantic importance rather than spatial organization. However, this improvement consequently fails to maintain the structured spatial layouts required by spatial architectures.

**Task-specific Token Reduction.** Recent token reduction techniques have increasingly targeted complex tasks, but many are designed for specific tasks or models, such as video understanding (Lee et al., 2024b; Shen et al., 2025; Choi et al., 2024), segmentation (Lu et al., 2023; Norouzi et al., 2024), or vision-language models (Ye et al., 2025; Hu et al., 2024; Alvar et al., 2025). While these methods have shown success in their specialized domains, they do not address the fundamental spatial compatibility challenge we tackle: maintaining structured token layouts essential for spatial architectures. A gap remains for general-purpose token reduction methods that can work effectively with the growing prevalence of spatial architectures.

## 3 METHODS

Existing token reduction methods face a fundamental dilemma: they either fail to preserve spatial structure or fail to exploit uneven information density across the spatial layout (see Figure 1). To

Figure 3: 2D token reduction with spatial-aware merging: (1) original 14×14 tokens, (2) select horizontal tokens to merge, (3) merge horizontally to 14×12 tokens, (4) select vertical tokens to merge, (5) merge vertically to 12×12 tokens.[1]

address this, we employ (i) a 2D token reduction strategy to enforce structured spatial layout, (ii) a spatial-aware token merging that selectively targets redundant tokens while preserving relative spatial relationships, and (iii) a max-magnitude-per-dimension token representation that preserves salient features without requiring layer-wise attention rescaling.

## 3.1 2D REDUCTION STRATEGY

We observe that existing token reduction methods break 2D spatial coherence due to uneven token counts across rows and columns (see Figure 1(b)). To address this, we reduce tokens in each dimension sequentially, to ensure consistent token counts per row and column.

To operate on $H \times W$ tokens representing a 2D spatial layout, the 2D reduction performs two sequential phases (illustrated in Figure 3):

1. **Horizontal Reduction:** Reduce $r_w$ tokens from each row, resulting in $H \times (W - r_w)$ tokens.
2. **Vertical Reduction:** Reduce $r_h$ tokens from each column, resulting in $(H - r_h) \times (W - r_w)$ tokens.

To better adapt to window attention, we perform 2D reduction independently within each window, restricting token merging among tokens within the same window. This is achieved by first partitioning the feature map into non-overlapping windows, then applying our 2D reduction algorithm to each window's token set independently.

Both phases use our spatial-aware token matching algorithm described in Section 3.2.

## 3.2 SPATIAL-AWARE TOKEN MATCHING

**Graph Construction.** Since our 2D reduction operates on each row and column independently, tokens within each subset naturally form a linear arrangement based on their spatial positions. Motivated by this, we use a **path graph** to further preserve spatial coherence, where each token only connects to its adjacent neighbors within the same row or column. This design ensures that merged tokens maintain the original relative spatial positions of their constituent tokens, which is critical for 2D positional embeddings (Li et al., 2022b; Heo et al., 2024). Additionally, path graphs minimize the number of edges by connecting only adjacent tokens, reducing the complexity of computing similarity scores from $O(N^2)$ to $O(N)$ compared to global token merging algorithms such as ToMe (Bolya et al., 2023) that computes all pairwise token similarities.

Our path graph construction enforces a strict adjacency constraint: tokens can only merge with their immediate spatial neighbors. This guarantees that the merged token's position maintains the original spatial ordering of its constituent tokens. In contrast, global matching approaches like ToMe (Bolya et al., 2023) merging across the entire token sequence: a token at position 0 may merge with one at position $W - 1$, with the resulting merged token placed at $(W - r_w) - 1$. Such long-range merging fundamentally disrupts the spatial correspondence between token positions and their original locations. While some spatially-aware methods like GTP-ViT (Xu et al., 2024) incorporate spatial proximity as a factor of consideration, they do not enforce adjacency as a hard constraint.

Consequently, these methods still permit merging between spatially distant tokens when similarity scores favor such pairings, failing to guarantee preservation of relative spatial relationship in the output token arrangement.

**Edge Selection.** The naive optimal approach would be to select the top-k most similar edges from the path graph for merging. However, this can create processing dependencies which limit parallelization when three or more adjacent tokens must be merged. For example, three adjacent tokens must be merged as either $((i, j), k)$ or $(i, (j, k))$, which requires two steps. In general, these dependency chains can grow, requiring either a linear or logarithmic number of steps, depending on the implementation.

To enable better parallelization, we adopt ToMe's (Bolya et al., 2023) node bipartition approach, which alternates token role assignments so that adjacent tokens have complementary roles (source and destination). Each source token then nominates its most similar adjacent neighbor as its merge destination, and we select the top-k edges from these nominations. This guarantees that no more than three tokens are ever merged, so merging never takes more than two steps. However this does not strictly guarantee the selection of the most similar edges.[2]

**Ablation Studies.** We conducted ablation experiments to evaluate the trade-off between parallelization efficiency and edge selection optimality. As shown in Table 1, comparing our bipartite approach ("bipartite, local") against naive top-k edge selection ("naive, local") reveals minimal performance differences, and the bipartite approach achieves same or better mIOU in 5 out of 8 experimental settings, while the parallelization enables higher speedups compared to the naive approach. Based on this, we adopt bipartite edge selection with path graph as our design. Additionally, we evaluate against global bipartite matching ("bipartite, global") from ToMe (Bolya et al., 2023). This comparison validates that our spatially-constrained approach outperform the conventional global matching.(See Table 1)

Table 1: Ablation studies comparing design choices for CubistMerge against alternative design choices and commonly used existing methods. Experiments were conducted on 500 randomly selected COCO training images with token merging methods applied off-the-shelf on SAM-H and SAM-B across different token reduction rates and application depths. Results show mIOU drop and speedup relative to the baseline model without token reduction. Our chosen design is highlighted .

| | Application Depth | 0 | | | | 1/4 | | 1/2 | |
| | $r_h = r_w$ | 4 | | 8 | | 4 | | 4 | |
| Method | | mIOU drop | Speedup | mIOU drop | Speedup | mIOU drop | Speedup | mIOU drop | Speedup |
|---|---|---|---|---|---|---|---|---|---|
| SAM-H | bipartite, local | -2.23% | 1.68 | -3.61% | 2.05 | **-1.47%** | 1.47 | **-0.72%** | 1.31 |
| | naive, local | **-2.17%** | 1.64 | **-3.53%** | 1.99 | -1.57% | 1.44 | -0.72% | 1.25 |
| | bipartite, global | -2.47% | 1.63 | -3.84% | 2.03 | -1.63% | 1.46 | -0.77% | 1.29 |
| SAM-B | bipartite, local | **-1.49%** | 1.69 | -2.48% | 1.94 | **-1.15%** | 1.44 | **-0.47%** | 1.27 |
| | naive, local | -1.60% | 1.55 | **-2.42%** | 1.70 | -1.16% | 1.36 | -0.51% | 1.15 |
| | bipartite, global | -1.62% | 1.69 | -2.59% | 1.95 | -1.16% | 1.43 | -0.51% | 1.27 |

(a) Graph construction and edge selection methods, discussed in Section 3.2

| | Application Depth | 0 | | | | 1/4 | | 1/2 | |
| | $r_h = r_w$ | 4 | | 8 | | 4 | | 4 | |
| Method | | mIOU drop | Speedup | mIOU drop | Speedup | mIOU drop | Speedup | mIOU drop | Speedup |
|---|---|---|---|---|---|---|---|---|---|
| SAM-H | Max-Per-Dim | **-2.23%** | 1.68 | **-3.61%** | 2.05 | **-1.47%** | 1.47 | **-0.73%** | 1.31 |
| | Max-Vector | -2.54% | 1.68 | -3.84% | 2.04 | -1.66% | 1.47 | -0.93% | 1.30 |
| | Weighted Average | -2.48% | 1.63 | -3.74% | 2.00 | -1.63% | 1.44 | -0.83% | 1.28 |
| SAM-B | Max-Per-Dim | **-1.49%** | 1.69 | **-2.48%** | 1.94 | **-1.15%** | 1.44 | -0.47% | 1.27 |
| | Max-Vector | -1.67% | 1.68 | -2.92% | 1.92 | -1.29% | 1.45 | -0.60% | 1.27 |
| | Weighted Average | -1.57% | 1.55 | -2.92% | 1.83 | -1.29% | 1.38 | **-0.44%** | 1.24 |

(b) Token representation methods, discussed in Section 3.3

## 3.3 Max-Magnitude-Per-Dimension Token Representation

Another key contribution lies in how we represent merged tokens. The most commonly used token representation is weighted averaging introduced in ToMe (Bolya et al., 2023), creating merged representations that are insufficient to attract appropriate attention for the multiple tokens they represent. ToMe addresses this using proportional attention scaling, which can introduce computational overhead of up to 3% of runtime for DeiT-B and complicating adoption in models with optimized attention implementations.

---

[2]See Section A.4 for visualization.

To address this issue, we observe that (i) high-magnitude values in token embeddings naturally attract more attention, reflecting more salient features; and (ii) averaging among multiple tokens reduces those highest-magnitude values. Instead of averaging, therefore, we perform max-magnitude-per-dimension operations across tokens being merged, preserving the most prominent values from each dimension. This eliminates the need for token size tracking and layer-by-layer attention scaling while ensuring merged tokens remain representative.

Formally, for a set of tokens $\{t_1, t_2, \ldots, t_n\}$ to be merged, where each token $t_j \in \mathbb{R}^d$, the merged token $t_m$ is computed as

$$t_m[i] = t_c[i] \text{ where } c = \underset{j \in \{1,2,\ldots,n\}}{\operatorname{argmax}} (|t_j[i]|)$$

for each dimension $i \in \{0, 1, 2, \ldots, d\}$. This operation selects, for each dimension independently, the value from whichever token has the maximum absolute value in that dimension, effectively preserving the most salient feature across all candidate tokens while maintaining both magnitude and sign information.

**Ablation Studies.** We compare our approach against two other methods: (1) weighted average, the most commonly used token representation method introduced in ToMe (Bolya et al., 2023), and (2) max-magnitude-vector, which selects the token with the highest L1 norm, serves to validate whether our method's superior performance stems from the per-dimension selection principle or merely from choosing more values from tokens that happen to be more informative. As shown in Table 1, per-dimension consistently outperforms both methods across different settings in both accuracy and speedup. This validates that our max-magnitude-per-dimension approach is genuinely superior to the standard averaging method and not merely benefiting from selecting tokens that are more suitable for preservation during merging.

# 4 EXPERIMENTS

We conduct comprehensive experiments across various architectures and vision tasks to evaluate the effectiveness of CubistMerge. Our experimental design addresses three primary research questions: (i) Does our method effectively preserve spatial relationships across diverse spatial architectures? (ii) How does our approach generalize across different vision tasks? and (iii) How does our method's performance compare against existing token reduction methods?

To answer the first question, we include spatial architectures with diverse spatial components in our evaluation. We include models that use both shifting (Liu et al., 2021) and non-shifting (Li et al., 2022a) window attention. For 2D positional embeddings, we include models that use decomposed relative positional embeddings (Li et al., 2022b) and RoPE (Heo et al., 2024). This diversity evaluates our method's compatibility across the spectrum of modern spatial architectural designs.

To address generalizability, we evaluate across diverse vision tasks including image classification, object detection, instance segmentation and panoptic segmentation.

To assess the competitiveness of our method, we conduct comprehensive comparisons against Expedite (Liang et al., 2022a), the only prior method capable of preserving spatial structure, across every spatial architecture experiment. To enable even broader comparative evaluation against a wider range of existing methods, we extend our evaluation to non-spatial architectures, where more existing methods sare compatible.

**Experiment Setup and Metrics.** We use performance metrics and datasets consistent with the original baseline models. Speedups are calculated from runtime measurements conducted on RTX 2080 Ti, except for DINOv3 experiments which were measured on V100. FLOPS are computed using the fvcore library (Meta Research, 2023). By default, all experiments apply token reduction methods off-the-shelf without additional training, with fine-tuning results specifically noted where applicable.

**Experiment Configuration.** The experiment configuration involves two key variables: the layer $l$ where token reduction is applied and the reduction rate. The reduction rate is specified by parameters $r_h$ and $r_w$, which denote how many tokens are reduced from every row and column respectively. Given $H \times W$ input tokens, we use integer values $r_h = m, r_w = n$ for models with consistent token counts across all inputs, resulting in $(H - m) \times (W - n)$ tokens. For models where token counts vary based

on input image size, we use fractional values $r_h = a, r_w = b$, resulting in $(H - a \times H) \times (W - b \times W)$ tokens.

## 4.1 Spatial Architectures

We evaluate across classification, object detection, instance segmentation and panoptic segmentation on models with spatial architectures including DINOv3 (Siméoni et al., 2025), MViTv2 (Li et al., 2022b), ViTDet (Li et al., 2022a), SAM (Kirillov et al., 2023), SAM2 (Ravi et al., 2025) and Mask2Former (Cheng et al., 2022).

**Prior Works Comparison.** We primarily compare against Expedite (Liang et al., 2022a) as it's the only existing method suitable for spatial architectures. Additionally, we include ToMe (Bolya et al., 2023) in selected experiments to demonstrate the performance gap between spatial-preserving and non-spatial-preserving methods. We exclude ToMe results when the performance degradation exceeds 20% as such large drops preclude meaningful comparison.

To assess layer sensitivity, we conduct experiments with token reduction inserted at different layers within each architecture, examining how performance varies when merging is applied at early vs. later layers. Throughout our experiments, we demonstrate that CubistMerge consistently outperforms existing methods while maintaining more consistent performance across different layers.

**Layer Selection.** We ensure all of our evaluations include results from Expedite's optimal configuration to guarantee fair comparison. When available, we use the recommended layer settings from the original Expedite paper or official repository. Otherwise, we systematically test Expedite across 4–6 different layers to identify its best-performing configuration. We ensure all selected layers maintain reasonable performance without substantial metric degradation. Additionally, we conduct layer sensitivity analysis using a default early layer configuration, typically the first layer, or for architectures with multiple stages, the first layer of the deepest stage.

### 4.1.1 Model Sweep

**MViTv2.** We evaluate MViTv2 (Li et al., 2022b), which uses decomposed relative positional embeddings, on image classification. Table 2 presents layer sensitivity results and includes fine-tuning results for MViTv2-B, with fine-tuning limited to 3 epochs. For meaningful comparisons, we use $l = 10$ for MViTv2-B and $l = 20$ for MViTv2-L, where Expedite and ToMe achieve more reasonable performance, and vary $r_h, r_w$ to produce the results shown in Figure 4.

**DINOv3.** We evaluate image classification and object detection using ViT7B backbone (Siméoni et al., 2025), which incorporates RoPE for 2D positional embeddings. Classification results are shown in Table 2 and Figure 4, while object detection results are shown in Table 3 and Figure 5. We experimented at $l = 10$ and $l = 20$, and selected $l = 20$ where Expedite and ToMe exhibits more reasonable results, for further experiments varying $r_h$ and $r_w$. Despite using the same pretrained backbone, ToMe and Expedite exhibit much worse layer sensitivity for object detection at $l = 10$, while CubistMerge maintains consistent performance across both tasks.

**ViTDet.** We further evaluate object detection using ViTDet (Li et al., 2022a), which employs window attention and decomposed relative positional embedding (Li et al., 2022b). We used the best performing backbone (ViT-H) with Mask R-CNN and Cascade Mask R-CNN as baseline. We apply CubistMerge with $l = 0$ by default. However, Expedite performs poorly at $l = 0$ with over 40 in AP drop, so we use $l = 2$ for Expedite (determined experimentally as the best performing configuration for Expedite). Results are shown in Figure 5.

**SAM.** We evaluate instance segmentation on SAM (Kirillov et al., 2023) which uses ViTDet backbone architecture. Figure 6(a) shows layer sensitivity analysis demonstrating CuMe's superior consistency across layers compared to Expedite. We conduct full evaluations on COCO (Figure 6(b)) and ADE20K (Figure 6(c)) across all model variants using bounding box prompts, with CuMe applied at $l = 0$ and Expedite at its recommended layer[3].

---

[3]Expedite only provided recommended settings ($l = 6$ and $l = 16$) for SAM-H. We scale the relative depth accordingly for SAM-L and SAM-B.

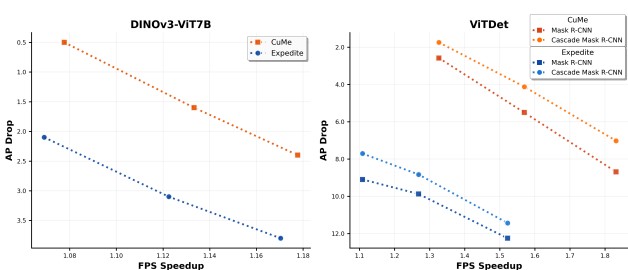

Figure 5: Object detection results. For DINOv3-ViT7B, we vary $r_h = r_w = 0.1, 0.15, 0.2$ at $l = 20$. For ViTDet, we vary $r_h = r_w = 4, 8, 12$ at $l = 0$ for CuMe and $l = 2$ for Expedite.

Table 3: Layer sensitivity analysis of object detection on DINOv3-ViT7B with $r_h = r_w = 0.1$

| Model | AP | Speedup |
|---|---|---|
| Baseline | 57.4 | 1.00× |
| ToMe $l=20$ | 52.9 | 1.08× |
| Expedite $l=20$ | 55.3 | 1.07× |
| CuMe $l=20$ | 56.9 | 1.08× |
| ToMe $l=10$ | 31.0 | 1.13× |
| Expedite $l=10$ | 37.2 | 1.11× |
| CuMe $l=10$ | 55.5 | 1.12× |

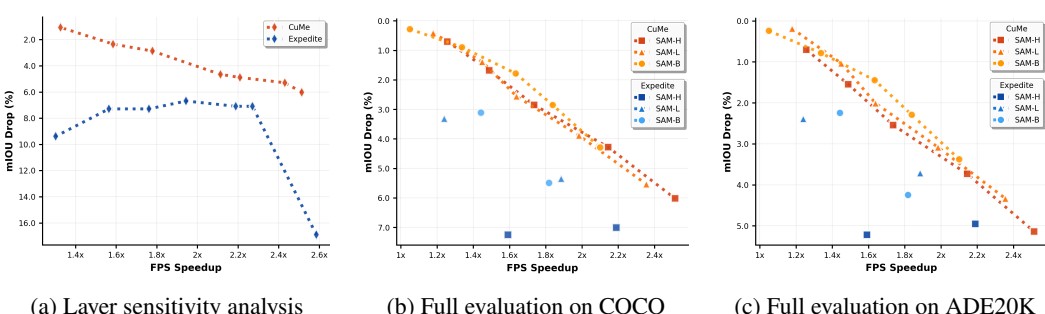

(a) Layer sensitivity analysis     (b) Full evaluation on COCO     (c) Full evaluation on ADE20K

Figure 6: Instance segmentation results on SAM. (a) shows sensitivity analysis by applying CuMe and Expedite at 7 different layers of SAM-H evaluated on COCO. (b) and (c) shows full evaluation on COCO and ADE20K, applying CuMe at layer $l = 0$ and Expedite at $l = 6$ and $l = 16$.

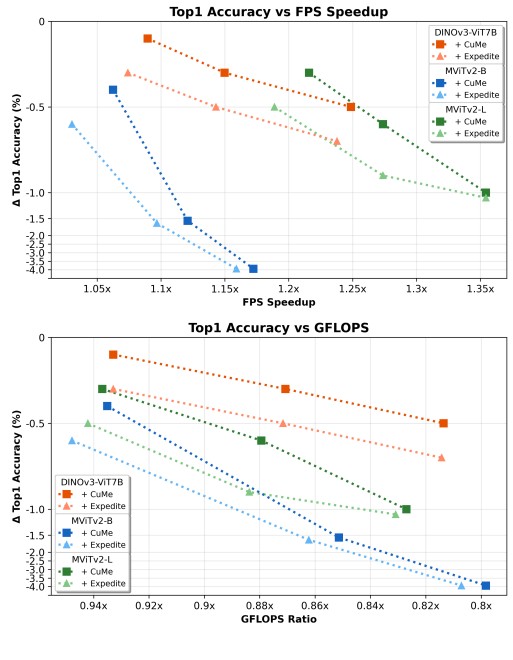

Figure 4: Image classification results on spatial architectures, varying $r_h = r_w = 1, 2, 3$ with $l = 10$ on MViTv2-B, $l = 20$ on DINOv3-ViT7B and MViTv2-L.

Table 2: Image classification results on spatial architectures, with $r_h = r_w = 1$. Fine-tuned results are included for MViTv2-B within 3 epochs of training.

| DINOv3-ViT7B | | | | |
|---|---|---|---|---|
| Method | Top1(%) | Top5(%) | Speedup | GFLOPS |
| Baseline | 88.0 | 98.4 | 1.00× | 1349.9 |
| ToMe $l=10$ | 84.3 | 97.1 | 1.12× | 1214.0 |
| Expedite $l=10$ | 87.1 | 98.2 | 1.12× | 1215.3 |
| CuMe $l=10$ | **87.7** | 98.2 | 1.12× | 1213.9 |
| ToMe $l=20$ | 86.9 | 98.1 | 1.07× | 1259.3 |
| Expedite $l=20$ | 87.7 | 98.4 | 1.07× | 1259.2 |
| CuMe $l=20$ | **87.9** | 98.4 | 1.09× | 1259.2 |
| **MViTv2-B** | | | | |
| Method | Top1(%) | Top5(%) | Speedup | GFLOPS |
| Baseline | 84.2 | 96.8 | 1.00× | 10.2 |
| ToMe $l=7$ | 69.8 | 88.3 | 1.07× | 9.3 |
| Expedite $l=7$ | 81.4 | 95.5 | 1.05× | 9.5 |
| CuMe $l=7$ | **82.6** | 96.2 | 1.07× | 9.3 |
| ToMe $l=10$ | 79.8 | 94.6 | 1.06× | 9.5 |
| Expedite $l=10$ | 83.6 | 96.5 | 1.03× | 9.6 |
| CuMe $l=10$ | **83.8** | 96.6 | 1.06× | 9.5 |
| Expedite $l=10$ | 83.8 | 96.7 | 1.03× | 9.6 |
| CuMe $l=10$ | **84.1** | 96.7 | 1.06× | 9.5 |
| **MViTv2-L** | | | | |
| Method | Top1(%) | Top5(%) | Speedup | GFLOPS |
| Baseline | 85.3 | 97.1 | 1.00× | 43.9 |
| ToMe $l=9$ | 68.5 | 87.6 | 1.24× | 39.8 |
| Expedite $l=9$ | 83.8 | 96.4 | 1.20× | 40.0 |
| CuMe $l=9$ | **84.3** | 96.6 | 1.26× | 39.8 |
| ToMe $l=20$ | 83.9 | 96.6 | 1.20× | 41.1 |
| Expedite $l=20$ | 84.8 | 97.0 | 1.19× | 41.3 |
| CuMe $l=20$ | **85.0** | 97.0 | 1.22× | 41.1 |

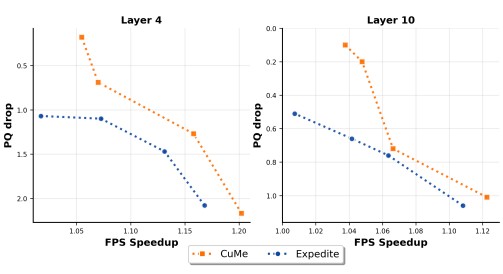 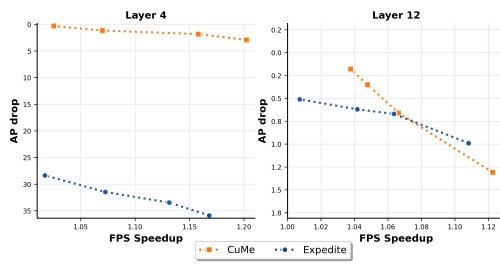

(a) Panoptic segmentation performance          (b) Instance segmentation performance

Figure 7: Results on Mask2Former(Swin-L), varying $r_h = r_w = 1, 2, 3$

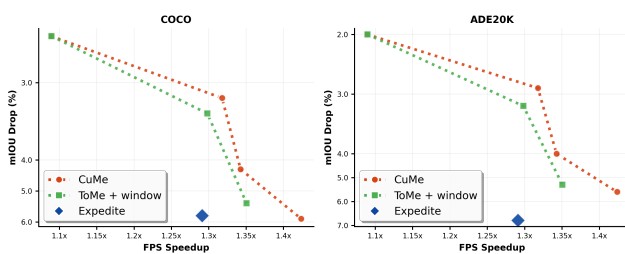

Figure 8: Instance segmentation results on SAM2-L evaluated on COCO and ADE20K datasets, varying $r_h = r_w = 4, 8, 12$ with CuMe and ToMe at $l = 9$, and Expedite at $l = 15$.

Table 4: Segmentation results on Segmenter(ViT-S), comparing CuMe against ALGM on Cityscapes and Pascal Context

| Method | mIOU | Speedup | GFLOPS |
|---|---|---|---|
| **Cityscapes** | | | |
| Baseline | 76.54 | 1.00× | 115.97 |
| ALGM | 75.24 | 1.75× | 61.56 |
| CuMe | 75.44 | 1.71× | 65.51 |
| **Pascal Context** | | | |
| Baseline | 53.01 | 1.00× | 32.09 |
| ALGM | 52.97 | 1.35× | 22.28 |
| CuMe | 52.95 | 1.27× | 23.43 |

**SAM2.** We extend our evaluation to SAM2 (Ravi et al., 2025), which uses Hiera (Ryali et al., 2023) backbone that employs window attention. Expedite generally performs poorly on SAM2 with 20+% mIOU drops; we experimented Expedite on 6 different layers and reported the best performance found. To provide additional baselines, we leverage the fact that SAM2 does not use 2D positional embeddings and has consistent window partition within its deepest stage, allowing ToMe to operate on each window individually to achieve reasonable results. We apply CuMe and ToMe at $l = 9$ (the first layer of the deepest stage). Figure 8 presents instance segmentation results on SAM2-L across COCO and ADE20K datasets.

**Mask2Former.** Mask2Former (Cheng et al., 2022) is an architecture capable of addressing any image segmentation task, with Swin-L (Liu et al., 2021) being its best performing backbone which we adopt for our evaluation. We evaluate CuMe against Expedite on both panoptic and instance segmentation tasks using the COCO dataset, with results presented in Figure 7. We conduct experiments at two layers: the layer from Expedite's recommended settings and at the first layer of Swin-L's deepest stage ($l = 4$).

Our method consistently outperforms Expedite and ToMe, often by significant margins, across all experiments conducted in this section, while showing superior consistency across different layers.

### 4.2 Non-Spatial Architectures

To enable comparison against more existing token reduction methods, we extend our evaluation to vanilla ViT without spatial components. In this section, we select models based on their established compatibility with existing methods: DeiT (Touvron et al., 2021) due to its foundational role and widespread adoption across token reduction literature, and Segmenter (Strudel et al., 2021) to enable comparison with ALGM (Norouzi et al., 2024) which targets segmentation tasks.

**DeiT** We evaluate on DeiT-B compared against ToMe (Bolya et al., 2023), PiToMe (Tran et al., 2024), ToFu (Kim et al., 2024), MCTF (Lee et al., 2024a), GTP-ViT (Xu et al., 2024) and DynamicViT (Rao et al., 2021). We evaluate both off-the-shelf and with fine-tuning limited to 5 epochs. Figure 9 shows results using existing methods' recommended merging schedules and CuMe applied at $l = 1$. Fine-

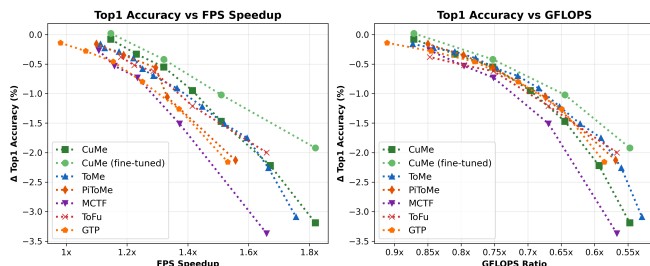

Figure 9: Image classification results on DeiT-B compared against prior token reduction methods.

Table 5: Fine-tuned results on DeiT-B, within 5 epochs.

| Fine-tuned Results | | | |
|---|---|---|---|
| Method | Top1 | Speedup | GFLOPS |
| Baseline | 81.80 | 1.00× | 17.58 |
| ToMe | 81.69 | 1.11× | 15.33 |
| PiToMe | 81.66 | 1.10× | 14.94 |
| ToFu | 81.59 | 1.18× | 14.89 |
| DyViT$_{0.9}$ | 81.83 | 1.10× | 15.53 |
| CuMe | 81.82 | 1.15× | 15.31 |
| MCTF | 80.96 | 1.34× | 9.93 |
| DyViT$_{0.7}$ | 81.44 | 1.33× | 11.49 |
| CuMe | 81.38 | 1.32× | 13.23 |

tuned results are shown in Table 5. CuMe achieves state-of-the-art results with no accuracy loss at 1.15× speedup within just one epoch of fine-tuning, while maintaining competitive performance across higher speedup ratios.

Interestingly, CuMe demonstrates superior speedups despite having slightly higher GFLOPS in some settings. We attribute this to computational overhead not captured by GFLOPS calculations: existing methods require attention scaling and token size tracking during inference, while our max-magnitude-per-dimension approach (Section 3.3) eliminates these overheads entirely. This observation is supported by our finding that fvcore GFLOPS measurements remain identical whether attention scaling is enabled or disabled, indicating that such runtime overheads are not reflected in theoretical GFLOP counts.

**Segmenter** We evaluate on Segmenter (Strudel et al., 2021) with ViT-S backbone compared against ALGM (Norouzi et al., 2024) on Cityscapes and Pascal Context datasets, applying both methods off-the-shelf. We adopt a merging schedule similar to ALGM's configuration. However, ALGM uses an adaptive method that automatically determines a similarity threshold for token merging, which we cannot directly adopt for our graph-based approach due to fundamental algorithmic differences. Instead, we apply CuMe at the same layers as ALGM ($l = 1$ and $l = 5$) with the same $r_h$ and $r_w$ values at both layers. Results are shown in Table 4.

Results demonstrate that CuMe achieves competitive performance against this broader range of existing methods on non-spatial architectures as well.

## 5 CONCLUSION

In this paper, we proposed CubistMerge, a novel token merging method that preserves spatial integrity through structured 2D reduction, spatial-aware merging, and max-magnitude-per-dimension representation. Extensive experiments demonstrate state-of-the-art performance and broad generalizability across diverse vision tasks and architectures.

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

## A  APPENDIX

### A.1  LLM USAGE

LLM was used only to polish writing at small scale (a few sentences).

### A.2  REPRODUCIBILITY

Code will be made available at an anonymous GitHub account: `https://github.com/0118-999-88999-9119-725-3`.

### A.3  CHALLENGES OF SPATIAL ARCHITECTURES

#### A.3.1  WINDOW ATTENTION

**Background.** The Swin Transformer (Liu et al., 2021) introduced sliding window attention to address the quadratic complexity of global self-attention in vanilla ViT (Dosovitskiy et al., 2021). By restricting self-attention to non-overlapping local windows, Swin achieves linear complexity while maintaining modeling capacity through shifted windowing for cross-window connections (see Figure 10). This mechanism has since become foundational in subsequent architectures. ViTDet (Li et al., 2022a) validated the effectiveness of window attention for dense prediction tasks, and showed that simpler window attention without shifting is sufficient when aided by a few cross-window propagation blocks. This non-shifting variant was then adopted by state-of-the-art models (Kirillov et al., 2023; Ravi et al., 2025; Ryali et al., 2023).

**Why Preserving Spatial Layout is Critical.** Window attention operates on local spatial regions, leveraging the high correlation of nearby visual features (Liu et al., 2021). This requires tokens to maintain a coherent 2D spatial arrangement to be partitioned into windows, otherwise tokens may lose opportunities to attend to tokens in local regions, as shown in Figure 10's demo of ToMe. The shifted windowing scheme further depends on this structured layout to enable cross-window connections.

Token reduction methods that fail to preserve spatial structure break this assumption: unstructured methods like ToMe (Bolya et al., 2023) produce irregular token layouts where different windows reduce varying numbers of tokens after merging. This leads to two unpalatable options: (1) maintaining the original window grouping — so that each window contains a different number of tokens — is at odds with the regular SIMD dataflow properties that accelerators like GPUs rely on for performance; while (2) padding all windows to the same length would offset the computation reduction benefit of token merging.

Another naive solution might be to reduce $H \times W$ tokens to $H' \times W'$ tokens and treat the reduced set as a new 2D token layout that can be partitioned into windows. This naive approach has also been used in LTM (Wang & Yang, 2025). However, as shown in Figure 10, this approach destroys the spatial correspondence between the original and reduced layouts, as windows may now group spatially distant tokens together while placing spatially local tokens in different windows, defeating the purpose of local window attention.

#### A.3.2  2D POSITIONAL EMBEDDING

**Background.** Decomposed relative positional embeddings were introduced in MViTv2 (Li et al., 2022b), encoding spatial relationships based on relative spatial distances between tokens. Unlike absolute positional embeddings in vanilla ViT (Dosovitskiy et al., 2021), these learned parameters are injected into *each attention layer* and computed *separately along height and width dimensions*. SAM adopts this strategy, combining it with window attention for strong zero-shot capabilities (Kirillov et al., 2023). RoPE (Heo et al., 2024), on the other hand, encodes 2D spatial relationships through axial frequency operations applied, again, separately for x and y dimensions. DINOv3 (Siméoni et al., 2025) demonstrates RoPE's effectiveness for self-supervised visual representation learning.

---

[4]See Section A.7 for details on attention visualization methodology.

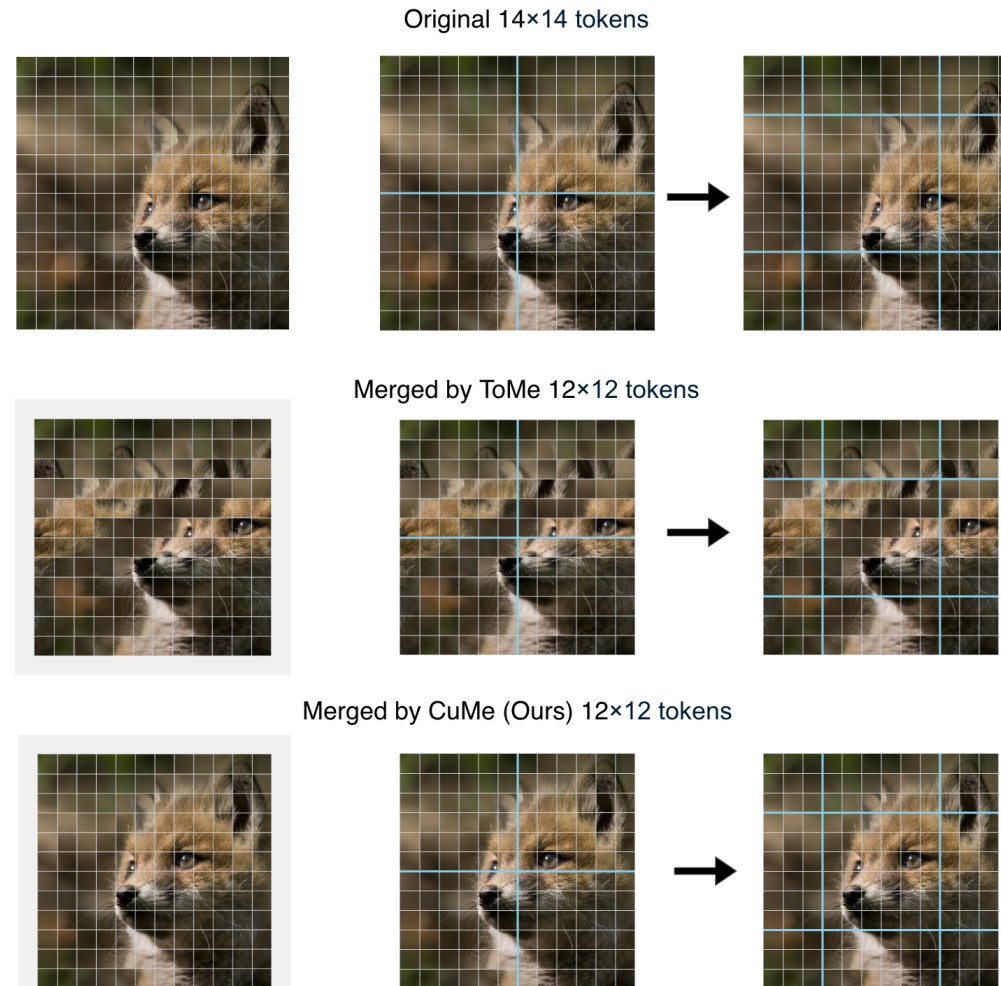

Figure 10: Window attention with shifted window partitioning. Cyan lines indicate window boundaries. Top row shows the original token layout with regular and shifted window partitioning. Middle and bottom rows demonstrate how token merging methods interact with window attention: ToMe (middle) destroys spatial structure, causing misalignment with window boundaries after merging, while CuMe (bottom) preserves the structured layout, maintaining compatibility with both regular and shifted window partitions.

**Why Preserving Spatial Layout is Critical.** 2D positional embeddings require tokens to maintain their relative spatial positions to correctly compute spatial relationships. When token reduction methods destroy the spatial structure, the positional embeddings can no longer accurately represent the spatial relationships between tokens, leading to significant performance degradation.

We demonstrate this with decomposed relative positional embeddings (Li et al., 2022b). As illustrated in Figure 11, positional bias (a) enhances attention from spatially near regions, as seen in the comparison between (b) and (c).

To understand the impact of token merging on spatial architectures that use positional embeddings, we visually compare attention patterns that appear in ToMe (Bolya et al., 2023) (non-spatial) and CuMe (spatial). For ToMe (d,e), we observe significant distortion from the baseline attention patterns, as the irregular token arrangement after merging disrupts the relative positional relationships that the embeddings depend on. In contrast, CuMe (f,g) maintains structured token layouts that preserve relative positions, and yields attention patterns that closely align with the baseline. This demonstrates why preserving spatial structure is critical for architectures relying on 2D positional embeddings.

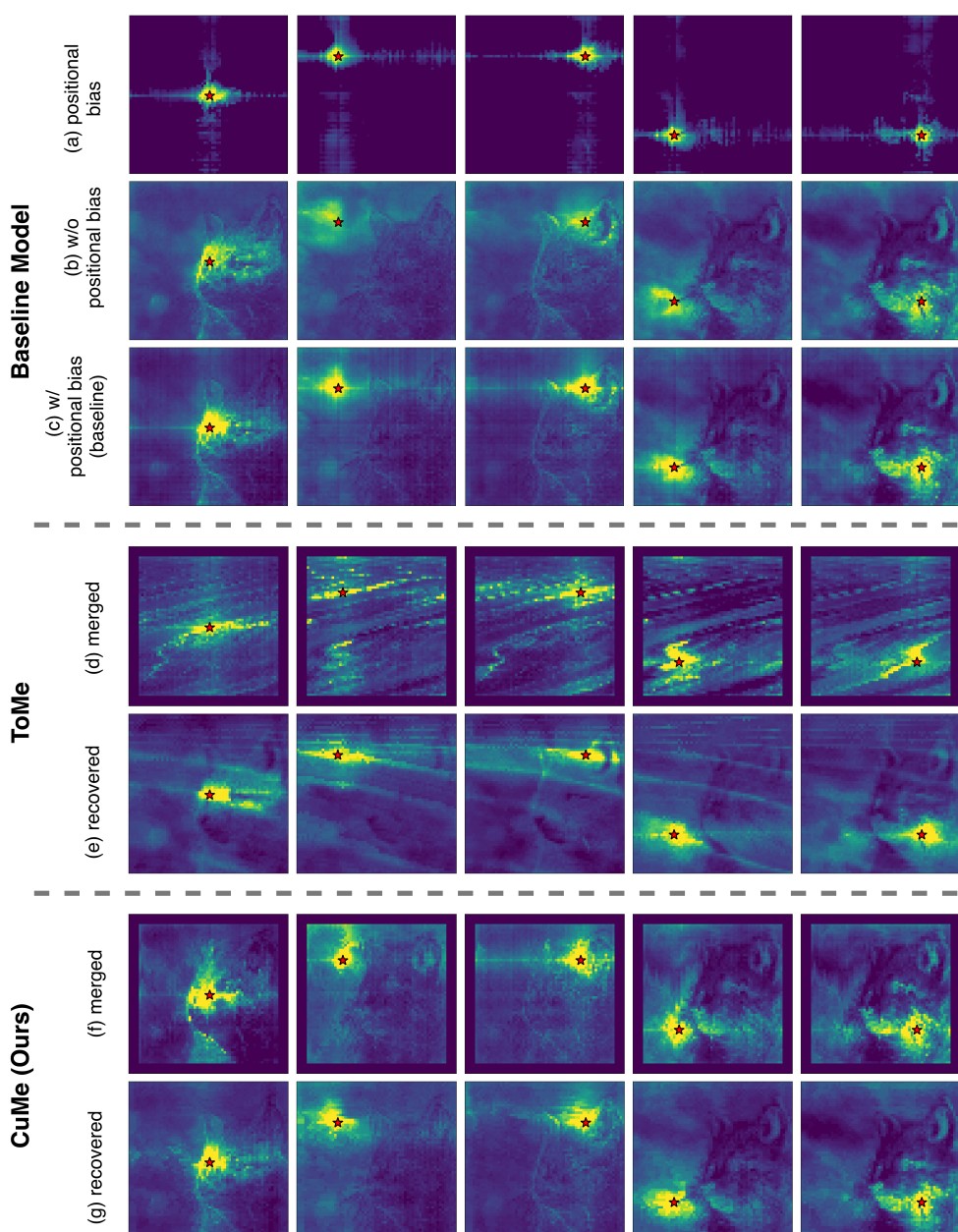

Figure 11: Visualization of relative positional embedding with attention heat map towards 5 different token positions indicated by red stars. Baseline (top): (a) positional bias component, (b) attention score without positional bias, and (c) attention score with positional bias (actual attention score). ToMe (middle): (d) attention score and (e) effective attention score on original spatial layout, recovered from attention score on merged layout. CuMe (bottom): (f) and (g) similarly for CuMe. Collected from block 2 of SAM-B.[4]

## A.4 Details on Graph Construction and Edge Selection

Figure 12 provides visualization of the path graph construction and edge selection approaches discussed in Section 3.2. Our path graph construction (Figure 12(a-c)) connects only spatially adjacent tokens within each row or column, preserving relative spatial relationships. This differs from ToMe's (Bolya et al., 2023) global bipartite matching (Figure 12(d)), which matches tokens

globally without spatial constraints; as demonstrated by our ablation results in Table 1, ToMe's global approach is less effective for spatial architectures.

**Naive Edge Selection.** The naive optimal approach would select the top-k most similar edges from the path graph. However, this creates processing dependencies when a token serves as both source and destination of selected edges. For example, if edges b→c, c→d, and d→e are selected, token c must first receive b before merging into d, forcing sequential execution with $O(N)$ time complexity (Figure 12(a)).

Reduction tree approaches can theoretically improve this to $O(\log N)$ by organizing merging operations (Figure 12(b)): tokens b→c and d→e merge simultaneously in the first step, then c→d is redirected to merge c→e in the second step. However, such tree-structured computations are not a good match for SIMD execution in GPUs, and are less well supported by frameworks like PyTorch. Moreover, even with logarithmic complexity, the sequential dependencies still limit parallelization.

**Bipartite Solution.** To resolve the complexity bottleneck, we adopt ToMe's parallelization technique through node bipartition. We alternate token role assignments so that adjacent tokens have complementary roles (source and destination). Each source token nominates its most similar adjacent neighbor, and we select the top-k edges from these nominations (Figure 12(c)). This guarantees that no token participates in more than one merge operation at each step, eliminating processing dependencies and enabling parallel execution with scatter-reduce operations. While this does not strictly guarantee selecting the most similar edges, Table 1 shows minimal performance differences while achieving better speedups.

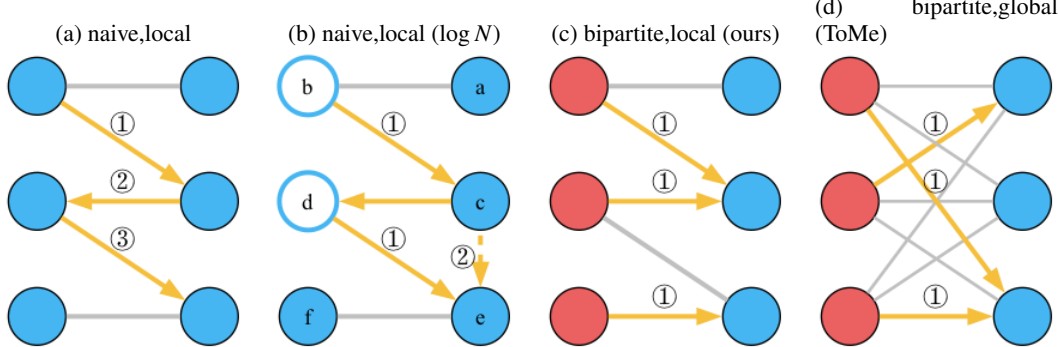

Figure 12: Illustration of edge selection algorithms. Arrows on selected edges (in orange) indicate the direction of token merging, pointing from the source token to the destination token. The numbers on edges represent execution order of merging required by dependencies. (a) Path graph with naive edge selection, requiring sequential execution. (b) Path graph with naive edge selection, optimized with reduction tree to $O(\log N)$ complexity. (c) Path graph with bipartite edge selection to eliminate dependencies by ensuring each source token (in red) can only merge to one destination (in blue). (d) Bipartite Soft Matching, the global bipartite matching approach from ToMe.

## A.5 DETAILS ON 2D REDUCTION IMPLEMENTATION

We implement the 2D reduction strategy described in Section 3.1 with explicit tensor operations below:

---

**Algorithm: 2D Token Reduction**

**Input:** $X \in \mathbb{R}^{N \times H \times W \times D}$
**Output:** $X'' \in \mathbb{R}^{N \times (H - r_h) \times (W - r_w) \times D}$

```
// Horizontal Reduction
```
$X = X.\text{transform}([N, H, W, D] \rightarrow [N \times H, W, D])$
$X' = \text{BatchTokenMerge}(X, r_w)$
```
// Vertical Reduction
```
$X' = X'.\text{transform}([N \times H, (W - r_w), D] \rightarrow [N \times (W - r_w), H, D])$
$X'' = \text{BatchTokenMerge}(X', r_h)$
**return** $X''.\text{transform}([N \times (W - r_w), (H - r_h), D] \rightarrow [N, (H - r_h), (W - r_w), D])$

---

The BatchTokenMerge operations in both phases reduce $r_w$ or $r_h$ tokens from each sample of the batch, using our spatial-aware token matching algorithm described in Section 3.2.

## A.6 Token Recovery for Dense Prediction Tasks

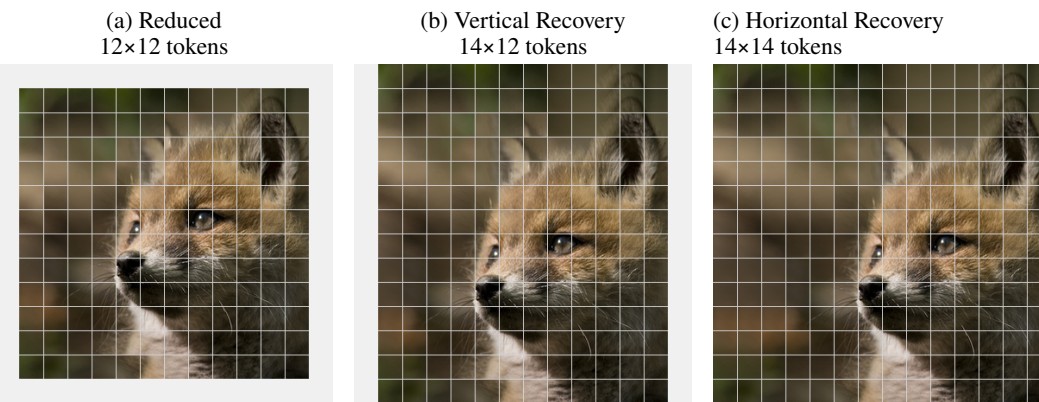

|(a) Reduced|(b) Vertical Recovery|(c) Horizontal Recovery|
|12×12 tokens|14×12 tokens|14×14 tokens|

Figure 13: Token recovery process for CubistMerge. Starting from (a) the reduced 12×12 layout, we (b) first recover the vertical dimension to 14×12, then (c) recover the horizontal dimension to the original 14×14 layout. Grid lines show token boundaries at each stage.

### A.6.1 2D Token Recovery

Token recovery is performed in reverse order of the reduction process: we first recover the vertical dimension, then the horizontal dimension, as illustrated in Figure 13.

### A.6.2 Token Recovery within Row/Column

We employ the simple token recovery that has been implicitly used in ToME (Bolya et al., 2023). Given multiple source tokens $\{x_{\text{src}_1}, x_{\text{src}_2}, \ldots, x_{\text{src}_k}\} \in \mathbb{R}^c$ that merge into a single destination token, we define:

$$x_{\text{merged}} = \text{MergeTokens}([x_{\text{src}_1}, x_{\text{src}_2}, \ldots, x_{\text{src}_k}]) \tag{1}$$

For recovery, each source token is recovered by duplicating the merged token:

$$x'_{\text{src}_i} = x_{\text{merged}} \quad \forall i \in \{1, 2, \ldots, k\} \tag{2}$$

Note that our MergeTokens operation uses max-magnitude-per-dimension representation (see Section 3.3), which differs from the weighted averaging used in ToME (Bolya et al., 2023).

## A.7 Details on Attention Rescaling / Proportional Attention

Following the recommendations from ToME (Bolya et al., 2023), we do not apply attention rescaling on MAE (He et al., 2022) pretrained models in all of our experiments, whether measuring ToME

performance or the weighted average experiment from our ablation studies in Table 1, since the model used (SAM (Kirillov et al., 2023)) is MAE pretrained. We still observed slight computational overhead in ablation experiments from Table 1 because token size tracking was still needed.

## A.8 DETAILS ON MEASUREMENTS OF PRIOR WORKS

All experimental results presented in this manuscript, including those of prior works, are from our own execution of the code and represent measurements that we have independently measured and verified. We conducted these experiments with scientific rigor and ethical responsibility, following best practices to ensure reproducibility and fairness:

- We always use the official implementation released by the original authors if provided.
- We always use the recommended environment and library versions specified by the authors if provided.
- We measure all experiments on the same NVIDIA GeForce RTX 2080 Ti hardware to ensure consistency, except for DINOv3 experiments which exceed the memory capacity of this GPU. For DINOv3 experiments, we use the NVIDIA Tesla V100 model borrowed from a shared compute cluster (we cannot guarantee the same physical hardware instance across all runs due to cluster scheduling).
- For wall clock runtime measurements, we always pre-heat the GPU by running a certain number of iterations before measurement to ensure that factors such as GPU temperature do not introduce inconsistencies.

Despite these efforts, some of our measurements cannot reproduce what the original papers reported.

### A.8.1 DISCREPANCIES IN ACCURACY MEASUREMENTS

In several cases, we found that the accuracy reported in the original papers is not reproducible, sometimes even with the code setup and evaluation scripts they provided. We also discovered instances where evaluation scripts used different crop rates or preprocessing parameters than those used for the baseline models, leading to inflated performance metrics. The accuracy results we report for prior works represent verified measurements produced under controlled and consistent experimental conditions, which may differ from originally published results due to corrected evaluation protocols.

### A.8.2 DISCREPANCIES IN GFLOPS MEASUREMENTS

To ensure fairness and consistency, all GFLOPs results presented in this manuscript are measured using the fvcore library (Meta Research, 2023), a state-of-the-art and commonly-used library for GFLOPs measurements. This provides a standardized and reproducible method for computing GFLOPS across all methods, allowing for accurate comparisons.

We cannot reproduce the GFLOPs counts reported in some papers. In some cases, the paper and released code do not include the details on how reported GFLOPS were calculated. In other cases, prior works' source code uses custom FLOP calculation code that differs from results measured with fvcore.

### A.8.3 DISCREPANCIES IN WALL-CLOCK TIME MEASUREMENTS

We report wall-clock time measured on NVIDIA GeForce RTX 2080 Ti, with the exception of DINOv3's results measured on V100. To ensure fairness, we measure all experiments (including baseline) on the same RTX 2080 Ti for all experiments except DINOv3. For DINOv3, we cannot guarantee using the same physical V100 instance across all runs due to cluster scheduling, but we ensured using the same V100 model. We always pre-heat the GPU by running a certain number of iterations before measurement to ensure that factors such as GPU temperature do not introduce inconsistencies. This allows us to provide consistent and reproducible runtime comparisons across all methods.

However, the absolute values may differ from originally published results in prior works: they may have used different hardware or CUDA versions. We are not able to reproduce some of the speedups reported in prior works.

