# OpenReview forum: "CubistMerge: Spatial-Preserving Token Merging For Diverse ViT Backbones"
_ICLR.cc/2026/Conference — Submitted to ICLR 2026_

### Official Review · Reviewer_EmTZ · 2025-10-25

**Soundness:** 2
**Presentation:** 1
**Contribution:** 2
**Rating:** 4
**Confidence:** 4

**Summary:**

The paper introduces a token merging method designed to preserve the 2D spatial structure of Vision Transformer (ViT) tokens during pruning.

**Strengths:**

The idea of perserving spatial relationship sounds novel and interesting for token pruning. Existing works rarely discuss this point.

**Weaknesses:**

However, while the core idea is strong, the current manuscript suffers from three major weaknesses:

Insufficient motivation in the Introduction: it does not clearly articulate why preserving spatial structure matters.
Lack of methodological clarity: key steps (e.g., how 2D reduction enforces uniform row/column counts) are described vaguely.
Experimental section reads like a report, not an argument—it presents numbers without synthesizing insights or addressing fundamental questions about downstream task compatibility.

**Questions:**

1. Why Preserve Spatial Structure matters? Modern ViT architectures increasingly embed explicit 2D spatial priors into their design, e.g., RoPE position embeddings. The paper assumes readers already understand why spatial coherence matters—but this is not justified.
2. Section 3 describes the approach at a high level but omits critical implementation details. After token merging, the feature map is smaller and sparser—how do segmentation/detection heads produce full-resolution outputs that align with input pixels?

---

> ### Author Response · Authors · 2025-11-28
> **Response to reviewer  EmTZ**
>
> We sincerely thank the reviewer for the constructive feedback.
>
> ## Q1 / W1: Why does preserving spatial structure matter?
>
> Spatial architectures rely on structured 2D token layouts for window attention and 2D positional embeddings.
>
> **Window Attention:** Window attention restricts self-attention to local spatial regions. When spatial coherence is broken, spatially distant tokens may be grouped together while local tokens are separated, substantially distorting the local window attention mechanism (see **Figure 10** in Appendix A.3).
>
> **2D Positional Embeddings:** 2D positional embeddings encode spatial bias on 2D token representations. Without spatial coherence, positional embeddings cannot function properly.
>
> In Appendix A.3, we added an analysis of decomposed relative positional embeddings (used in SAM) and illustrated the effects in **Figure 11**:
>
> - (top) how the 2D positional bias enhances attention from spatially near tokens,
>
> - (middle) how unstructured token merging (ToMe) distorts attention patterns, and
>
> - (bottom) how our spatial preserving approach preserves more baseline-aligned attention patterns.
>
> Unlike vanilla ViT, where positional embeddings are added only once at the beginning, these 2D positional embeddings are injected into _each attention layer_ (see Appendix A.3 for more details). This means we cannot simply apply unstructured merging after initial positional encoding is added as in vanilla ViT — spatial coherence must be maintained throughout all layers to ensure correct positional bias computation at each layer. For more details, please see our discussion with Reviewer u9e4 on Q3 on why we can't merge positional embedding and instead must keep feature tokens spatially coherent.
>
> For comprehensive analysis and detailed visualizations, see Appendix A.3.
>
> ## Q2: How do segmentation/detection heads produce full-resolution outputs that align with input pixels?
>
> For dense prediction tasks like segmentation and detection, we need to recover the reduced tokens back to the original spatial dimensions before feeding them to task-specific heads, which requires full resolution token layouts. We provide a detailed explanation of this recovery process in **Appendix A.6** (Token Recovery for Dense Prediction Tasks), which we summarize below.
>
> **2D Token Recovery:** Token recovery is performed in reverse order of the reduction process: we first recover the vertical dimension, and then the horizontal dimension (illustrated in Figure 12 in the appendix).
>
> **Token Recovery within Row/Column:** Within each row/column, we employ a simple token recovery approach: given multiple source tokens that were merged into a single destination token, we recover each source token by duplicating the merged token. This was also implicitly used in ToMe on a global basis, instead of within row/column.
>
> The recovered tokens can then be fed to segmentation or detection heads that expect full-resolution spatial layouts, ensuring proper pixel-level alignment for dense prediction outputs.
>
> Please see Appendix A.6 for more details.
>
> ## W2: Methodological clarity on 2D reduction
>
> Thank you for requesting clarification on our 2D reduction approach. We have added detailed implementation details in **Appendix A.5** (Details on 2D Reduction Implementation).
>
> ## W3: Experimental section reads like a report, not an argument
> Thank you for the feedback. We will improve the writing on experimental section to provide more insights with the results.

---

### Official Review · Reviewer_u9e4 · 2025-10-30

**Soundness:** 4
**Presentation:** 4
**Contribution:** 3
**Rating:** 6
**Confidence:** 5

**Summary:**

This paper proposes CubistMerge, which is a spatial-aware token merging method. In contrast to existing token reduction solutions that result in uneven token counts across rows and columns, CubistMerge introduces separate reductions for columns (horizontal reduction) and rows (vertical reduction). As a consequence, CubistMerge always produces even token reduction and preserves the original spatial relationships among remaining tokens.

**Strengths:**

* This paper explicitly studies the spatial relationship preservation during token merging, which is a novel topic in token reduction, and can serve for hierarchical ViTs that utilizes window-based attention.

* The experiments are thorough. Notably, adopting DINOv3 and SAM as the backbones is very rare in token reduction methods.

**Weaknesses:**

* The overall novelty is insufficient. The main contributation of this work is the spatial-aware token matches strategy, which is a trivial extension to existing token merging methods. And, the even token reduction per row and column has been proposed in LTM [1] and implicitly used in some other works.

[1] Wang, Yancheng, and Yingzhen Yang. "Efficient visual transformer by learnable token merging." TPAMI, 2025.

**Questions:**

Questions:

* Can the authors provide visualization comparisons between ToMe's and CubistMerge's merged results?

* How does CubistMerge merge relative positional embeddings?

* What if merging tokens without preserving spatial relationship but merging their relative positional embeddings together?

Suggestions:

* In the Graph Construction subsection, GTP-ViT should be mentioned as it also generates graphs on the image tokens to preserve spatial information on token reduction, while RoPE should be removed since it does not closely align with this topic.

* Performance on standard backbones, such as DeiTs and Swins is better to be provided (maybe in the appendix) for a fundamental comparison to existing approaches.

---

> ### Author Response · Authors · 2025-11-28
> **Response to Weakness 1**
>
> We sincerely thank the reviewer for the thoughtful and constructive feedback.
>
> ## W1: Novelty concern regarding overlap with LTM
>
> Thank you for bringing LTM to our attention. After reviewing the literature and inspecting their source code, we verified that their approach differs significantly from our per-row and per-column reduction strategy. The key difference is visualized in Figure 1(b) versus Figure 1(d).
>
> **LTM's approach:** LTM reduces $H×W$ tokens to $H'×W'$ tokens, then reshapes them to a 2D representation $[H', W']$ (with $H' = H - r_h, W' = W - r_w$). This reduction happens globally without spatial restrictions, and is the same naive approach we used when applying ToMe to spatial architectures in Section 4.1, visualized in Figure 1(b).
>
> LTM's approach has two key aspects that can cause significant changes in spatial relationships among tokens:
>
> 1. **No spatial restriction on merging**: Using $[h, w]$ to represent token position, a token at $[0, 0]$ may merge with a token at $[H-1, W-1]$. The resulting merged token may end up as far as $[H', W']$, even though one of its source tokens was originally at $[0, 0]$.
>
> 2. **No enforcement of consistent reduced token counts per row/column**: Consider two neighboring tokens $t_1$ at $[0, 0]$ and $t_2$ at $[1, 0]$. Even if both tokens did not participate in merging, if $r_w + 1$ tokens are reduced from the first row, $t_2$ will end up at $[0, H'-1]$, placing the two originally-adjacent tokens at distance $H'-1$ apart.
>
> These two factors can cause tokens to deviate from their original neighbors by distances up to $\Delta h$ = H'-1 and $\Delta w$ = W'-1.
>
> **Our approach:** In contrast, CubistMerge restricts merging to adjacent neighbors within each row and column, and enforces consistent token counts _within each row and column_, as shown in Figure 1(d). The maximum distance a token can deviate from its neighbor post-merging is $\Delta h$ = $r_h$ and $\Delta w$ = $r_w$.

---

> > ### Author Response · Authors · 2025-11-28
> > **Response to Q1, Q2, Q3 and suggestions**
> >
> > ## Q1: Can the authors provide visualization comparisons between ToMe's and CubistMerge's merged results?
> >
> > Yes, please see Figure 1, which shows token layouts after merging for ToMe (Figure 1(b)) versus CubistMerge (Figure 1(d)).
> >
> > Additionally, we have added visualizations demonstrating how spatial architectures operate on merged tokens in Figures 9 and 10. See Appendix A.3 for details.
> >
> > ## Q2: How does CubistMerge merge relative positional embeddings?
> >
> > Great question. Since we _preserve a spatially coherent layout_ — that is, the relative positions of tokens with respect to one another remains close compared to their original layout — we do not merge positional embeddings. Instead, we apply positional embeddings for [H', W'] directly to our merged token layout.
> >
> > This works because both decomposed relative positional embeddings and RoPE are not fixed-size parameters, but are computed _dynamically_ given the H and W dimensions. After merging from [H, W] to [H', W'], the positional embedding mechanism simply re-computes embeddings for the reduced dimensions [H', W'] and applies them to the spatially coherent [H', W'] token layout.
> >
> > ## Q3: What if merging tokens without preserving spatial relationship but merging their relative positional embeddings together?
> >
> > This is an excellent question. We actually explored this option during our development.
> >
> > The key challenge is that 2D positional embeddings are injected at every attention layer. This means positional embedding merging would need to occur at each layer, and would introduce substantial computational overhead.
> >
> > While this approach does yield better model performance under the same token reduction rate, the added runtime overhead makes it significantly slower than our chosen approach. We provide experimental comparison below:
> >
> > | r | Ours || Merge Pos Emb ||
> > |---|---------|---------|---------|---------|
> > |   | mIOU | Runtime | mIOU | Runtime(s) |
> > | 4 | 0.8009 | 0.58 | 0.8028 | 0.60 |
> > | 8 | 0.7911 | 0.49 | 0.7949 | 0.542 |
> > | 12 | 0.7794 | 0.42 | 0.7840 | 0.487 |
> > | 16 | 0.7651 | 0.34 | 0.7698 | 0.432 |
> > | 20 | 0.7477 | 0.29 | 0.7530 | 0.40 |
> > | 24 | 0.7295 | 0.23 | 0.7331 | 0.3562 |
> >
> > We will add a section to appendix detailing the options we explored and experiments conducted.
> >
> > **Why positional embedding merging is needed at every layer in this alternative approach:**
> >
> > - **Decomposed relative positional embeddings**: These are trained separately at each attention layer, meaning they are different across layers and therefore requires merging at each layer.
> >
> > - **RoPE**: While RoPE parameters are consistent across layers and it is technically possible to merge RoPE once, it would require
> >
> >   - holding $B \times H \times W$ RoPE tensors instead of $H \times W$ in memory — since each sample in the batch has a different merge pattern — causing significant memory overhead, and
> >
> >   - changing the implementation to work with $B \times H \times W$ RoPE tensors, which is a less efficient implementation that requires notably more FLOPs.
> >
> > In either case, merging positional embeddings is not ideal.
> >
> > ## S1: In Graph Construction subsection, GTP-ViT should be mentioned.
> >
> > Thank you for this suggestion. We will add a discussion of GTP-ViT to the Graph Construction subsection in the manuscript. We have also added a quantitative comparison against GTP-ViT in Section 4.2.
> >
> > ## S2: Performance on standard backbones (DeiTs and Swins) should be provided for fundamental comparison to existing approaches
> >
> > Please see Section 4.2 for DeiT-B results, where we compare against 5 other existing approaches. We are currently working on Swin experiments and will add them to the manuscript.

---

### Official Review · Reviewer_24b5 · 2025-10-30

**Soundness:** 3
**Presentation:** 3
**Contribution:** 2
**Rating:** 2
**Confidence:** 5

**Summary:**

The authors proposed a token merging techniques that split the merge process by two separate vertical and horizontal merge steps. Selected tokens are then merged by max-pooling on each dimension. The selection process for edges follows that of ToMe's.

**Strengths:**

- The paper is straightforward and easy to read
- The merge method maintains spatial coherence of the image tokens after merging
- The experimental results are competitive against baseline methods (ToMe and Expedite)

**Weaknesses:**

- Limited novelty: the method follows closely the approach proposed by ToMe and Expedite, with the 2D merge being the only technical contribution
- Lack of comprehensive evaluation or more recent baseline: the experiments compare CuMe with ToMe and Expedite in the main results, and both methods were proposed in 2022. The submission cites GTP-ViT but claims that it requires extensive retraining, while GTP-ViT is actually a train-free method that consistently outperforms ToMe w/o fine-tuning.
- The technique advantage of having "spatial coherence" after merging needs further articulation. ViTs see an image in tokens, and the attention happens between tokens. If would be great for the authors to provide some insights about why the proposed method would help visual reasoning - is it because of positional embedding? I'm not able to get a good intuition of how this links to the ViT computations and translates to an advantage (from a ViT's PoV).

**Questions:**

See Weaknesses

---

> ### Author Response · Authors · 2025-11-28
> **Response to Weakness 1**
>
> We thank the reviewer for the detailed feedback. We address the raised concerns below.
>
> ## W1: Limited novelty - the method follows closely the approach proposed by ToMe and Expedite, with the 2D merge being the only technical contribution
>
> We respectfully disagree with this characterization. Our method is fundamentally different from Expedite, and while ToMe serves as a foundational work, we introduce substantial novelty to address the practical problem of spatial architecture compatibility.
>
> **Versus Expedite:** Our approach differs fundamentally in how similar tokens are identified.
>
> Expedite is **clustering-based**, a fundamentally different merging approach from the graph-based selective merging used in CuMe: Expedite employs a k-means clustering approach on superpixels initialized through adaptive average pooling.
>
> This results in CuMe being able to isolate and preserve semantically distinct tokens, where Expedite tends to cluster them with other tokens. For more details, see **Graph-based Token Merging** vs **Clustering-based Token Merging** in Section 2 (Related Works).
>
> We extensively compare against Expedite throughout our experiments not because we follow their approach, but because it is _the only existing method that preserves spatial structure_ — this makes it the only suitable baseline for evaluating spatial-preserving token reduction.
>
> **Versus ToMe:** While ToMe established the foundation for graph-based token merging, we employ a fundamentally different graph construction that targets spatial architecture compatibility. The path graphs in CuMe connect only spatially adjacent tokens, while ToMe's complete bipartite graph considers all pairwise token similarities.
>
> From ToMe, we adopted **only** the parallelization strategy (node bipartition), a technique that has been widely adopted in subsequent graph-based token reduction works such as MCTF [1], Token Fusion [2], and PiToMe [3]. The spatial compatibility problem we solve remains unaddressed by ToMe and its derivatives.
>
>
> ## References
>
> [1] Lee, Sanghyeok, et al. "Multi-Criteria Token Fusion with One-Step-Ahead Attention for Efficient Vision Transformers." CVPR, 2024.
>
> [2] Kim, Minchul, et al. "Token Fusion: Bridging the Gap between Token Pruning and Token Merging." WACV, 2024.
>
> [3] Tran, Chau, et al. "Accelerating Transformers with Spectrum-Preserving Token Merging." NeurIPS, 2024.

---

> ### Author Response · Authors · 2025-11-28
> **Response to Weakness 2 and 3**
>
> ## W2: Lack of comprehensive evaluation or more recent baseline
>
> **GTP-ViT:** Thank you for pointing out our misunderstanding of GTP-ViT [4]. We have conducted a more thorough literature review and updated our Related Works (Section 2) accordingly.
>
> Additionally, we have added a quantitative comparison against GTP-ViT in Section 4.2 (Non-Spatial Architectures), where we evaluate on DeiT-B against GTP-ViT along with 3 other more recent token reduction methods.
>
> **Baseline selection in spatial architecture experiments:** In these experiments (Section 4.1), we can only reasonably make an apples-to-apples comparison against Expedite, because it is the only other method that _preserves spatial structure_, which is critical for maintaining performance on spatial architectures (SAM, SAM2, DINOv3, Hiera, MViTv2, ViTDet, Mask2Former, etc.). See our response for W3 below for why spatial-preserving is critical for such architectures.
>
> All other methods (including ToMe, GTP-ViT, etc.) _lose structural relationships_, which causes significant performance drops on such spatial architectures, often exceeding 20% metric drops. We include ToMe _only_ to demonstrate this fact; we explain this in Section 4.1:
>
> > "We primarily compare against Expedite as it's the only existing method suitable for spatial architectures. Additionally, we include ToMe in selected experiments to demonstrate the performance gap between spatial-preserving and non-spatial-preserving methods. We exclude ToMe results when the performance degradation exceeds 20% as such large drops preclude meaningful comparison."
>
> Because spatial preservation is fundamental, _any_ method that does not preserve spatial structure will suffer significant performance degradation; it is sufficient to demonstrate this on one of them.
>
> **Broader baseline comparison:** To comprehensively evaluate against recent methods, including GTP-ViT, MCTF, Token Fusion, and ALGM [5], we included experiments on _non-spatial_ architectures for meaningful comparison. Please see Section 4.2 where we compare their performance on DeiT-B. This demonstrates that our method achieves competitive or state-of-the-art performance not only on spatial architectures but also on vanilla ViT backbones.
>
> ### W3: The technique advantage of having "spatial coherence" after merging needs further articulation
>
> Spatial architectures rely on structured 2D token layouts for window attention and 2D positional embeddings.
>
> **Window Attention:** Window attention restricts self-attention to local spatial regions. When spatial coherence is broken, spatially distant tokens may be grouped together while local tokens are separated, distorting the local window attention mechanism (see **Figure 10** in Appendix A.3).
>
> **2D Positional Embeddings:** 2D positional embeddings encode spatial bias on 2D token representations. Without spatial coherence, positional embeddings cannot function properly.
>
> In Appendix A.3, we added analysis on decomposed relative positional embeddings (used in SAM) and show visualization in **Figure 11**:
> - (top) how the 2D positional bias enhances attention from spatially near tokens
> - (middle) how unstructured token merging (ToMe) distorts attention patterns
> - (bottom) how our spatial preserving approach preserves more baseline-aligned attention patterns
>
> Note that unlike vanilla ViT, where positional embeddings are added only once at the beginning, these 2D positional embeddings are injected into _each attention layer_ (see Appendix A.3 for more details). This means we cannot simply apply unstructured merging after initial positional encoding is added as in vanilla ViT — spatial coherence must be maintained throughout all layers to ensure correct positional bias computation at each layer. For more details, please see our discussion with Reviewer u9e4 on Q3 on why we can't merge positional embedding and instead must keep feature tokens spatially coherent.
>
> For comprehensive background and detailed visualizations, see Appendix A.3.
>
> ## References
>
> [4] Xu, Xuwei, et al. "GTP-ViT: Efficient Vision Transformers via Graph-based Token Propagation." WACV, 2024.
>
> [5] Norouzi, Narges, et al. "ALGM: Adaptive Local-then-Global Token Merging for Efficient Semantic Segmentation with Plain Vision Transformers." CVPR, 2024.

---

### Official Review · Reviewer_u2DH · 2025-10-31

**Soundness:** 3
**Presentation:** 4
**Contribution:** 2
**Rating:** 8
**Confidence:** 4

**Summary:**

The authors proposed a set of ideas to reduce the number of tokens processed in transformer based vision networks. This task of reducing the number of tokens has been explored previously to reduce the computational requirements (both time and floating point operations) of such networks. Critically, the authors identified that current methods do not preserve spatial consistency after token reduction. The proposed method maintains spatial consistency and demonstrates superior performance as compared to previous method at the same level of token reduction. This is achieved by using a separable 2D reduction strategy, followed by merging only neighboring tokens, and finally a novel token merging scheme.

**Strengths:**

The presentation of the problem and the proposed method is intuitive, clear, and easy to follow. Ablation studies are provided for each of the claimed contributions. Furthermore, the authors tested their method extensively on numerous architectures, demonstrating that their proposed token reduction strategy can be applied to different architectures. Finally, the authors also tested their idea on non-spatial architectures for where the proposed components still make sense.

**Weaknesses:**

No significant weaknesses. Claims and contributions are supported by ablation studies and a rather thorough architecture sweep was performed.
Minor comments:

- There’s no need for paragraphs in the abstract, it can be made more concise too. Focus on the contributions and main results.
- Frames per second might be an interesting metric to show

**Questions:**

- What is the hardware used in assessing performance speedup?
- For deployment in real-world systems, especially robots or edge systems, where and how each step is performed might be of interest.

---

> ### Author Response · Authors · 2025-11-28
> **Response to Reviewer u2DH**
>
> We sincerely thank the reviewer for the thoughtful and constructive feedback. Your recognition of our work's is very encouraging.
>
> ## Response to Questions
>
> ### Q1: What is the hardware used in assessing performance speedup?
>
> Thank you for asking for this clarification.
>
> Please see Section 4 (Experiments) Experiment Setup and Metrics:
>
> > "Speedups are calculated from runtime measurements conducted on **RTX 2080 Ti**, except for DINOv3 experiments which were measured on **V100**."
>
> We use RTX 2080 Ti for all experiments except the experiments involving DINOv3. For DINOv3, we used V100 across all configs (baseline, CuMe, ToMe, Expedite) because DINOv3 memory requirements exceed the capacity of the RTX 2080 Ti. Note that we report _relative_ speedups, always computed from experiments running on the same hardware (i.e., both RTX 2080 Ti or both V100).
>
> ### Q2: For deployment in real-world systems, especially robots or edge systems, where and how each step is performed might be of interest.
>
> Because of the size and computational complexity of state-of-the-art ViT models, a GPU (or anther neural network accelerator) would likely be needed to run inference at reasonable frame rates, even with token reduction.
>
> CubistMerge therefore optimizes for parallel hardware in several ways:
>
> 1. **Graph construction (token similarity computation)** uses standard matrix operations, well-supported on GPUs.
>
> 2. **Bipartite edge selection** described in Section 3.2 uses scatter-reduce operations to run efficiently on GPUs.
>
> 3. **Top-k edge selection** uses the underlying hardware support, which is exposed and well-optimized in frameworks like PyTorch.
>
> ## Response to Minor Comments
>
> ### Abstract formatting and conciseness
> Thank you for this suggestion. We will revise the abstract to be more concise.
>
> ### Frames per second (FPS) metric
> Thank you for this suggestion, we will add FPS as an additional metric to the experiment section.
>
> The rationale for our choice of primarily using speedup is that speedup generalizes across different hardware, while absolute metrics (FPS and runtime) are hardware-dependent. Speedup also captures the same improvement for both runtime and FPS as they are inversely proportional (FPS = total runtime / total frames).

---

### Author Response · Authors · 2025-11-28
**Rebuttal Update from Authors**

We sincerely thank all reviewers for your thoughtful and constructive feedback. We have addressed all weaknesses and questions raised. We are currently working on further revisions to incorporate your suggestions and minor comments into the manuscript.

  ## Summary of Paper Revisions

  - Corrected categorization of GTP-ViT in Related Works (Section 2)
  - Added GTP-ViT experiments and comparison in experiments on non-spatial architectures (Section 4.2)
  - Added Appendix A.3: Background on spatial architectures and the importance of preserving spatial coherence
  - Added Appendix A.4: Graph construction and edge selection details
  - Added Appendix A.5: 2D reduction implementation details
  - Added Appendix A.6: Token recovery for dense prediction tasks details
  - Added Appendix A.8: Performance measurement methodology for prior works

We are currently working on additional changes to address minor comments and suggestions, and will update the manuscript before the end of the discussion period.

## Work-in-Progress Revisions

  - Abstract improvements (Reviewer u2DH)
  - FPS metric addition to experiments (Reviewer u2DH)
  - GTP-ViT discussion in Graph Construction subsection (Reviewer u9e4)
  - Performance evaluation on Swin backbone (Reviewer u9e4)
  - Enhanced experimental section with deeper insights (Reviewer EmTZ)

Best,
Authors

---

### Author Response · Authors · 2025-12-04
**Meta-Summary of Rebuttal**

We thank all reviewers for their feedback. Below, we list the key questions raised in the reviews, and summarize our responses.

## Key questions

### 1. Why does preserving spatial structure matter? (Reviewers 24b5, EmTZ)

**Response:** Spatial architectures require structured token layouts due to (1) **window attention** that operates on local spatial regions, and (2) **2D positional embeddings** injected at every attention layer. Without taking spatial coherence into account during token merging, a token may focus its attention on very different tokens than in the uncompressed version. We added **Appendix A.3** with visualizations (Figures 10–11) that show how non-spatial merging distorts attention patterns. See responses to Reviewer 24b5 (W3) and EmTZ (Q1) for details.

### 2. Novelty (Reviewers 24b5, u9e4)

**Question:** How closely does CubistMerge (CuMe) resemble ToMe and Expedite? Is the 2D merge the only technical contribution?

**Response:** Our core novelty lies in the stepwise 2D reduction and graph construction approach, which maintains spatial architecture compatibility while naturally preserving semantically significant tokens. In contrast, Expedite preserves the spatial architecture by clustering equal-area regions of the feature map; while this preserves spatial structure, it assumes equal information density and _does not isolate semantically significant tokens_.

ToMe, meanwhile, merges tokens _without regard to their relative spatial position_; as a result, the relative position of tokens is distorted (see Fig. 1b in the paper), which makes ToMe incompatible with modern spatially structured ViTs. The only element we adopt from ToMe is its parallelization strategy, which has been widely adopted among token merging works [1] [2] [3].

See responses to Reviewer 24b5 (W1) for more details.

**Question:** Has token reduction per row/column already been proposed in LTM?

**Response:** No. While LTM's token reduction also turns the original $H×W$ image into a $H'×W'$ image, it merely _reinterprets_ the reduced tokens as such; unlike CuMe, it _does not enforce consistent token counts per row/column during reduction_. As a result, tokens can move farther away from where they started, which more significantly changes the attention paid to those tokens compared to the original image. The per-row/column token reduction in CuMe (Section 3.1) is a novel way of constructing a spatially-preserving token array that stays more faithful to the original uncompressed attention patterns. See our response to Reviewer u9e4 (W1) for detailed technical comparison.

### 3. Did we compare against the most recent token reduction methods? (Reviewer 24b5)

**Response:** Yes, we do compare against the recent methods in **Section 4.2**, where we experiment on _non-spatial_ architectures. The question may stem from the _spatial_ architecture experiments in **Section 4.1**, where we compare primarily against Expedite because it is _the only existing method that preserves spatial structure_. As we explain in Section 4.1 of the original manuscript:

> We primarily compare against Expedite as it's the only existing method suitable for spatial architectures. Additionally, we include ToMe in selected experiments to demonstrate the performance gap between spatial-preserving and non-spatial-preserving methods.

See response to Reviewer 24b5 (W2) for details.

## Other Improvements
- Added GTP-ViT for comparison on non-spatial architecture experiments
- Added detailed explanation of motivation, and implementation details in appendix
- Some writing improvements

## References
[1] Lee, Sanghyeok, et al. "Multi-Criteria Token Fusion with One-Step-Ahead Attention for Efficient Vision Transformers." CVPR, 2024.

[2] Kim, Minchul, et al. "Token Fusion: Bridging the Gap between Token Pruning and Token Merging." WACV, 2024.

[3] Tran, Chau, et al. "Accelerating Transformers with Spectrum-Preserving Token Merging." NeurIPS, 2024.

---

### Meta-Review · Area_Chair_jjKG · 2025-12-29

**Summary:**

In summary, the reviewers pointed out weaknesses focus mainly on perceived limited novelty relative to prior token merging methods, and on initial shortcomings in motivation and clarity. Reviewer 24b5 argues that there is “limited novelty: the method follows closely the approach proposed by ToMe and Expedite, with the 2D merge being the only technical contribution,” and also criticizes “lack of comprehensive evaluation or more recent baseline,” pointing out that e.g. GTP-ViT is a train‑free method that “consistently outperforms ToMe w/o fine-tuning.” Reviewer u9e4 similarly states that “the overall novelty is insufficient,” describing the main contribution as “a trivial extension to existing token merging methods,” and claimfs that “even token reduction per row and column has been proposed in LTM and implicitly used in some other works.” Reviewer EmTZ notes that “the current manuscript suffers from three major weaknesses”: “insufficient motivation in the Introduction” regarding why spatial structure matters, “lack of methodological clarity” on key steps, and an experimental section that “reads like a report, not an argument.” Reviewer 24b5 also questions whether the advantage of “spatial coherence” is adequately articulated from a ViT’s point of view, asking for stronger intuition on why it improves visual reasoning.

Overall, the AC recommends rejection because of the overall reviewer opinion and the remaining novelty concerns suggest the work, while solid and practically useful, does not quite reach the threshold for ICLR. There is one clearly positive review (u2DH) and one mildly positive (u9e4), both emphasizing paper clarity, thorough experiments, and the practical importance of spatial‑preserving merging for SAM/DINOv3‑style architectures. These reviewer's assessments on their own are not specific enough to push the paper over acceptance. On the other hand, reviewer EmTZ and especially Reviewer 24b5 are unconvinced, characterizing the contribution as incremental, closely related to existing merging strategies, and only modestly differentiated from LTM and GTP-ViT. The rebuttal improves the paper by adding detailes, newer baselines, and stronger explanations of spatial coherence and dense‑task recovery, but much of this remains an extra, rather than reshaping the core contribution, and the main conceptual contribution can still be reasonably viewed as an engineering refinement of known ideas rather than a qualitatively new method. Given the split between one strong accept, one marginal accept, one marginal reject, and one firm reject with high confidence, given that strong doubts concern originality with the most positive reviewers not giving very specific reasons to accept, the AC does not see a compelling basis to overrule the more critical reviewer opinions.

On balance, after reading the paper, reviewer assessment, rebuttal/discussions, the AC sees no strong enough basis to recommend acceptance. The paper offers a well‑implemented, spatial‑preserving token merging scheme that performs well on SAM, DINO, and standard ViTs, with clear motivation and clarifications about window attention and 2D positional embeddings, but two reviewers still view the novelty over ToMe, Expedite, LTM, and GTP-ViT as limited, and the empirical narrative, even with new rebuttal additions, does not fully convince the AC over the impression of an incremental refinement rather than a substantially new idea/contribution that has a meningful impact on the community. The AC highly recommends the authors to address the concerns of the reviewers and take into account their suggestions of improvement when preparing a revised version.

**Reviewer Concerns:**

Some elements were missing in the original submission; e.g. There was no clear comparison to GTP-ViT and other recent reduction methods in the non‑spatial setting. The connection to LTM and whether even row/column reduction had already been explored was not fully clarified. Motivation for preserving spatial layouts, especially in relation to window attention and 2D positional embeddings across layers, was not explained in depth, and implementation details for 2D reduction and recovery for dense prediction heads were only sketched. The empirical narrative, as EmTZ notes, did not yet synthesize insights about why the method helps downstream tasks beyond showing numbers.

The rebuttal addresses a good fraction of the concerns. To Reviewer 24b5 and Reviewer u9e4, the authors argue that their approach differs fundamentally from Expedite (clustering‑based superpixels) and from ToMe (global bipartite merging), emphasizing their graph construction that restricts merging to spatial neighbors and enforces consistent reduced token counts per row/column. They contend that LTM “merely reinterprets” reduced tokens as a 2D grid without enforcing per‑row/column consistency during reduction, and thus allows larger spatial drift than CubistMerge; they explfain this with the Figure 1(b/d) comparison and distance arguments. They also correct their characterization of GTP-ViT and add it as a baseline in non‑spatial experiments on DeiT‑B (Section 4.2). To Reviewer EmTZ and Reviewer 24b5, they add Appendix A.3, which explains why preserving spatial structure matters for window attention and decomposed 2D positional embeddings in SAM‑like and DINOv3‑like architectures, and include visualizations (Figures 10–11) showing how unstructured merging distorts attention patterns and how CubistMerge better preserves them. To Reviewer u9e4 and Reviewer EmTZ, they add details on 2D reduction (Appendix A.5) and on token recovery for dense prediction tasks (Appendix A.6), describing the reverse merge and duplication strategy. They also discuss and empirically compare an alternative that merges positional embeddings at each layer, arguing it is slower for modest gains. These changes meaningfully strengthen the motivation and technical clarity, and broaden the evaluation with more recent baselines on non‑spatial ViTs.

At the same time, some concerns remain only partially resolved. Reviewer 24b5’s core view that the method is “limited novelty” and closely follows ToMe/Expedite is countered by the authors’ arguments, but the underlying structure—graph‑based merging with ToMe‑style parallelization, combined with a spatial constraint—can still reasofnably be seen as an incremental elaboration rather than a fundamentally new token‑reduction paradigm. Reviewer u9e4’s claim that the main contribution is “a trivial extension” and that row/column‑wise even reduction is close to ideaas in LTM may not be fully overturned by the rebuttal’s distinctions in distance bounds. Reviewer EmTZ’s criticism that the experimental section reads more like a report suggesfts that, even with added appendices, the main text still under‑argues the empirical story. Also, while the authors add GTP-ViT and mention plans for Swin experiments, the Swin results appear not yet integrated, and the novelty relative to other spatially‑aware graph‑based methods such as GTP-ViT is clarified but not decisively differentiated.

**Reviewer Scores:**

Because reviewers could not update their scores after the rebuttal, we can only speculate possible shifts/changes of their ratings. Reviewer u2DH, already sees no significant weaknesses; so the rebuttal’s extra details on hardware and deployment likely reinforce this stance. Reviewer u9e4, at rating 6: marginally above the acceptance thraeshold. But would not mind if paper is rejected, might appreciate the clarifications distinguishing CubistMerge from LTM and GTP-ViT and the new visuaalizations, but their novelty concern is focused and strongly stated, so a major rating change is uncertain. Reviewer EmTZ, at rating 4, marginally below the acceptance threshold, buut would not mind if paper is accepted, sees the core idea as strong but the manuscript as lacking in motivation and clarity; the added appendices and planned improvements to the experimental section address some of this, yet much of the new content sits outside the main narrative, so they probably remain cautious. Reviewer 24b5, at rating 2: reject, not good enough, with “5: absolutely certain” is unlikely to move significantly, as their main critique is about conceptual originality relative to ToMe, Expedite, and LTM, which the authors rebut but do not decisively/completely neutralize.

---

### Decision · Program_Chairs · 2026-01-26

Reject